# Real-World Cost-Effectiveness Analysis: How Much Uncertainty Is in the Results?

Heather K. Barr [1], Andrea M. Guggenbickler [1], Jeffrey S. Hoch [1,2,3,*] and Carolyn S. Dewa [1,4]

[1] Graduate Group in Public Health Sciences, Department of Public Health Sciences, University of California, Davis, CA 95616, USA

[2] Division of Health Policy and Management, Department of Public Health Sciences, University of California, Davis, CA 95616, USA

[3] Center for Healthcare Policy and Research, University of California, Davis, CA 95616, USA

[4] Department of Psychiatry and Behavioral Sciences, University of California, Sacramento, CA 95817, USA

[*] Correspondence: jshoch@ucdavis.edu

**Abstract:** Cost-effectiveness analyses of new cancer treatments in real-world settings (e.g., post-clinical trials) inform healthcare decision makers about their healthcare investments for patient populations. The results of these analyses are often, though not always, presented with statistical uncertainty. This paper identifies five ways to characterize statistical uncertainty: (1) a 95% confidence interval (CI) for the incremental cost-effectiveness ratio (ICER); (2) a 95% CI for the incremental net benefit (INB); (3) an INB by willingness-to-pay (WTP) plot; (4) a cost-effectiveness acceptability curve (CEAC); and (5) a cost-effectiveness scatterplot. It also explores their usage in 22 articles previously identified by a rapid review of real-world cost effectiveness of novel cancer treatments. Seventy-seven percent these articles presented uncertainty results. The majority those papers (59%) used administrative data to inform their analyses while the remaining were conducted using models. Cost-effectiveness scatterplots were the most commonly used method (34.3%), with 40% indicating high levels of statistical uncertainty, suggesting the possibility of a qualitatively different result from the estimate given. Understanding the necessity for and the meaning of uncertainty in real-world cost-effectiveness analysis will strengthen knowledge translation efforts to improve patient outcomes in an efficient manner.

**Keywords:** cost effectiveness; cancer interventions; real-world interventions; cancer; economic evaluation; healthcare; statistics; uncertainty

## 1. Introduction

Real-world cost-effectiveness analyses of new cancer treatments and interventions provide valuable evidence that can inform and potentially recalibrate healthcare decisions [1]. Initial funding decisions for expensive cancer drugs can be based on trial data or, more commonly, decision models based on data and assumptions informed by previous research [2]. The clinical trial setting, however, is heavily controlled, and the population being studied is often a restricted sub-population [3]. After a treatment has been covered by healthcare payers (e.g., the Ministry of Health), based on trial results, data accrue on the real-world costs and effectiveness, making the examination of real-world cost effectiveness in the general population feasible. Real-world results can be compared with initial estimates of the extra cost and extra effect to assess the actual efficiency in healthcare investments. When making or revisiting a decision on which treatments should be funded, results from real-world studies can be used to assist decision makers interested in attaining the most health from their limited healthcare budgets.

A previous rapid review of the existing literature explored real-world cost-effectiveness analyses of new cancer treatments and interventions in Canada [4]. The authors examined

cost, effect, and cost-effectiveness statistics. Whether these estimates came from administrative datasets or were from decision models, statistical estimates are still estimates and should be accompanied by methods for expressing statistical uncertainty [5]. Methods to characterize uncertainty are an important way to communicate the strength of the evidence that a new treatment is cost-effective. Indeed, prominent health economists recommend that the analytic focus on cost effectiveness "should be on the estimation of the joint density of cost and effect differences, the quantification of uncertainty surrounding the incremental cost-effectiveness ratio and the presentation..." [6]. Thus, it is important to study statistical uncertainty in real-world cost effectiveness and to make recommendations about how this crucial information is being communicated to healthcare decision makers.

We introduce five ways to characterize statistical uncertainty in cost-effectiveness analysis and report patterns about their use in published work identified through a rapid review of real-world cost-effectiveness analyses of cancer treatments and interventions in Canada. Subsequently, we make recommendations based on these findings and conclude with important next steps for those interested in communicating the real-world value of novel cancer treatments to medical professionals and policy makers accurately by incorporating information about the statistical uncertainty surrounding the results.

### 1.1. Background
Cost-Effectiveness Estimates

Using hypothetical data, Table 1 calculates the two main statistics that summarize cost-effectiveness analysis (CEA) results. The first is the incremental cost-effectiveness ratio (ICER). The ICER is the ratio of the extra cost over the extra effect. For example, if the extra cost ($\Delta$C) of a new treatment is USD 7980.19 and the extra effect is 0.02293 years (about 8.34 days), then the ICER equals approximately USD 348,024 per additional year of life. It is important to remember that an ICER of nearly USD 350,000 per additional year of life does not mean that the new treatment costs USD 350,000, nor does it provide one more year of life. If one pays USD 7980.19 more for an additional 0.02293 years of life, one is spending at a rate of nearly USD 350,000 per extra year of life. Thus, the ICER is a rate per 1 unit of outcome. Whether the ICER estimate represents good value for money depends on one's willingness to pay (WTP) for an additional unit of outcome (e.g., one more year of life).

**Table 1.** Estimates of Cost, Effect, and Cost Effectiveness from a hypothetical dataset.

| Group Variable | Mean | 95% Confidence Interval |
|---|---|---|
| Usual care, *n* = 49 | | |
| Cost | USD 292.1837 | (82.77, 501.60) |
| Effect | 0.7036735 | (0.6719546, 0.7353924) |
| New treatment, *n* = 53 | | |
| Cost | USD 8272.3770 | (5747.35, 10,797.40) |
| Effect | 0.7266038 | (0.6937985, 0.7594091) |
| Incremental | | |
| Cost | USD 7980.19 | (5375.24, 10,585.15) |
| Effect | 0.02293 | (0.02227, 0.06813) |
| Main statistics | | |
| Cost-effectiveness ratio (ICER) | USD 348,024 | (−372,322, 102,390) |
| Net benefit (INB) * | USD 45 | (−16,455, 16,546) |

Note: *n*, size of assumed sample. Willingness to pay (WTP) is assumed to be USD 350,000 in this hypothetical example. * The INB is calculated as (USD 350,000 × 0.02293) − USD 7980.19 ≈ USD 45.

As an alternative statistic in cost-effectiveness analysis, the incremental net benefit (INB) compares the actual value of what one gains in relation to the additional costs. The INB expresses just how cost effective a new treatment is because it incorporates the willingness-to-pay value. The actual value of 0.02293 more years of life is related to how much one is willing to pay for an additional year of life. If a healthcare payer is willing to pay USD 350,000 for an extra year of life, then a gain of 0.02293 more years of life provides additional benefits worth USD 8025.50 (i.e., USD 350,000 × 0.02293). These additional benefits (worth USD 8025.50) come at an extra cost of USD 7980.19. Thus, the INB (i.e., the difference between the additional benefits and the additional costs) is about USD 45. When the INB estimate is positive (e.g., USD 45 > 0), a new treatment is considered cost-effective, since the estimate of the extra benefits outweighs the estimate of the extra costs.

### 1.2. Five Common Ways to Characterize Statistical Uncertainty in a CEA

In the following section, we review five ways commonly used to characterize statistical uncertainty in a cost-effectiveness analysis. We begin with the ICER 95% Confidence Interval (CI) and then describe the INB 95% CI, the scatterplot on the cost-effectiveness plane, the Cost-Effectiveness Acceptability Curve (CEAC), and the INB by WTP plot.

#### 1.2.1. Computing an ICER's 95% Confidence Interval

Estimates of the ICER, like other statistical estimates, require a description of their statistical uncertainty. A common way to characterize statistical uncertainty for estimates is by reporting a 95% confidence interval (95% CI). Methods to compute 95% CIs for ICERs were developed in the late 1990s and early 2000s, but evidence of the dissemination of these methods is not readily evident in published papers in clinical journals. The rightmost column in Table 1 provides examples of 95% CIs for the ICER. A major drawback of the 95% CI for the ICER is that, sometimes, it cannot be computed [7] and, other times, it appears non-sensical. For example, the 95% CI for the ICER is (−372,322 to 102,390) in Table 1. While the 95% CI does appear to include 0, it does not appear to include the ICER estimate.

#### 1.2.2. Computing an INB's 95% Confidence Interval

Like with the ICER, a common way to characterize the INB's statistical uncertainty is by reporting a 95% CI. Methods to compute 95% CIs for the INB were developed in the late 1990s and early 2000s. For the INB, the non-statistically significant confidence interval means we cannot rule out that the extra benefits are no different from the extra costs. For the ICER, the negative values presage critical statistical challenges and potentially incorrect interpretation. A negative ICER is difficult to interpret without information on which cost-effectiveness quadrant the values or estimates lie. For example, does the negative value come from cost savings ($\Delta C < 0$) or from a lower effectiveness ($\Delta E < 0$) [8]?

#### 1.2.3. Scatterplot on the Cost-Effectiveness Plane

It is possible to illustrate the ICER on the cost-effectiveness plane, which has extra cost on the vertical axis and extra effect on the horizontal axis. At point (0,0), there is equal cost and effect to usual care or the comparison group. The ICER estimate is the slope of the ray from the origin to the point at ($\Delta E$, $\Delta C$), represented by "R", where $\Delta E$ is the estimate of the extra effect and $\Delta C$ is the estimate of the extra cost. In this case, the ICER equals the slope of the dashed line in Figure 1A that extends from (0,0) to (0.02293, USD 7980.19).

The ICER's 95% CI is centered around the ICER estimate, which means that the 95% CI (−372,322, 102,390) encloses the dashed line. Figure 1B illustrates this and clarifies how it is possible for an estimate of USD 348,024 to be at the center of a 95% CI equal to (−372,322, 102,390). The solid lines represent the upper and lower 95% confidence limits with slopes equal to −372,322 (for the left solid line) and 102,390 (for the right solid line).

The method used to construct the 95% CI for the ICER, based on Fieller's theorem, involves making parametric assumptions (e.g., the $\Delta E$ and the $\Delta C$ are jointly normally

distributed) [9]. Very skewed data or very small sample sizes tend to weaken beliefs that the central limit theorem (i.e., assuming normality is ok) applies, and methods that "mimic" other types of distributions are frequently employed (e.g., bootstrapping or probabilistic sensitivity analysis). These methods produce 1000s of simulated results, which can be plotted in a scatterplot to illustrate uncertainty. Figure 1C shows the results of such a simulation, plotting the different results from running the study 1000s of times.

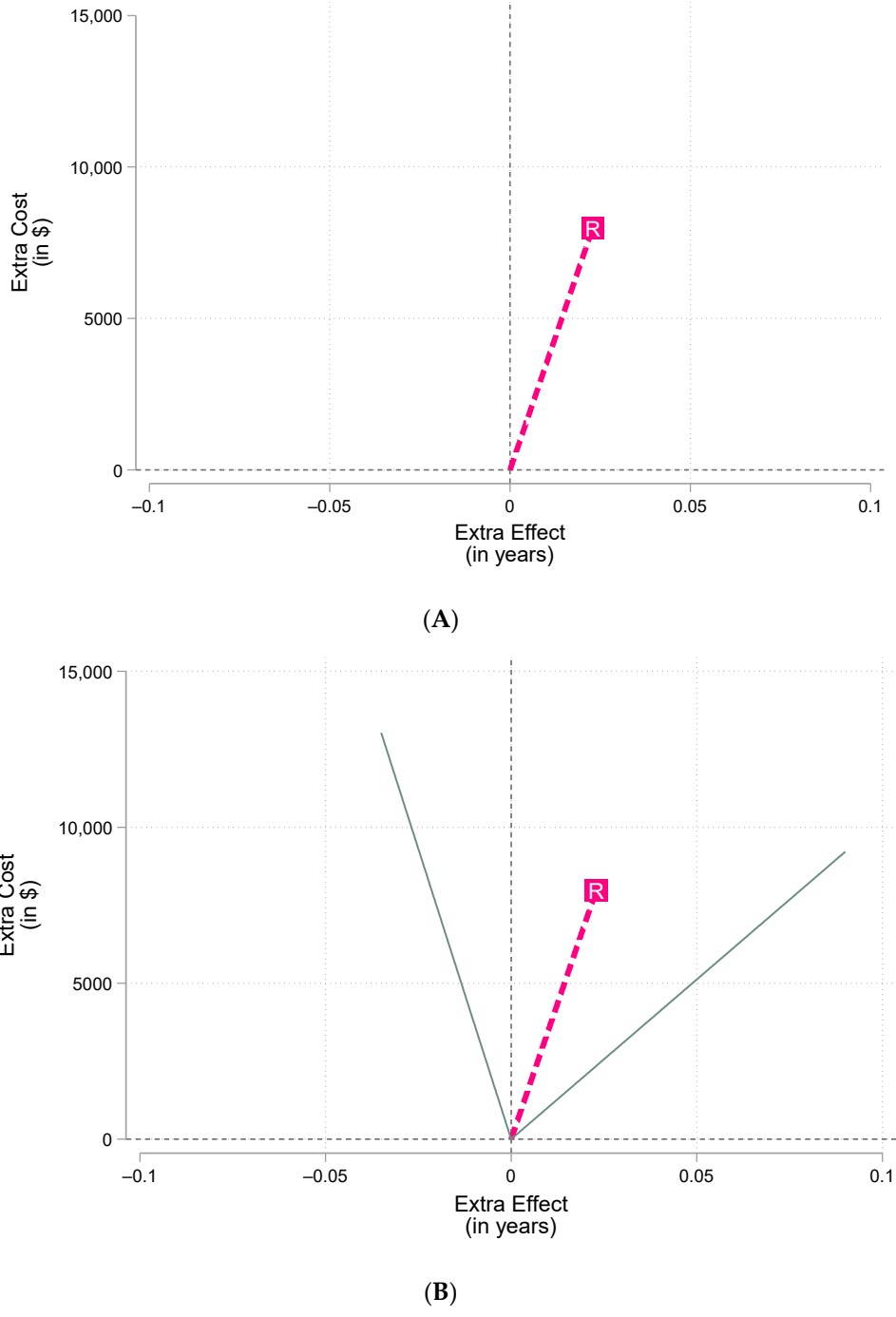

**(A)**

**(B)**

**Figure 1.** *Cont.*

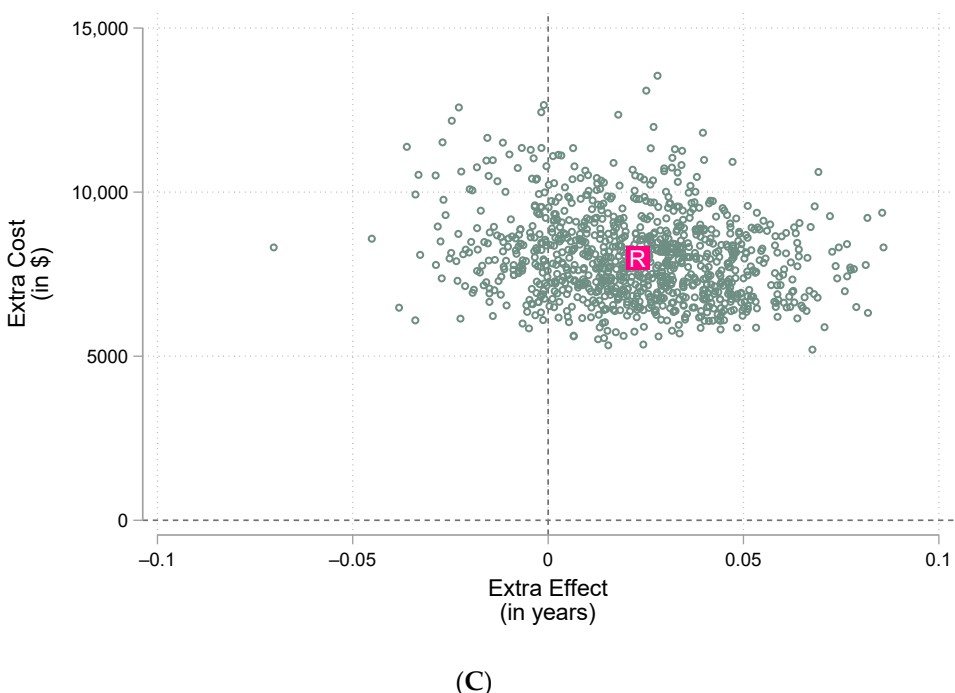

(**C**)

**Figure 1.** (**A**) Incremental cost-effectiveness ratio (ICER) illustrated on a cost-effectiveness plane. (**B**) The ICER and its 95% confidence interval illustrated on a cost-effectiveness plane. (**C**) The scatterplot approach to showing ICER uncertainty on a cost-effectiveness plane.

Figure 1A–C all show the same estimate of the couplet ΔE and ΔC (it is marked with an "R" in the figures). The figures differ in how they convey uncertainty. Figure 1A does not show any uncertainty; only the ICER estimate is presented (as the slope of the dashed line from the origin). Given the uncertain nature of the ICER estimate (i.e., it is either based on one dataset and/or assumptions and "expert opinion"), it is essential to characterize the statistical uncertainty as well. Figure 1B shows the 95% CI (as solid lines emanating from the origin), assuming joint normality of ΔE and ΔC. Figure 1C makes different "non-parametric" assumptions about the distribution of ΔE and ΔC.

The fact that Figure 1C has "dots" (data points representing extra cost and extra effect estimates) that are all above the horizontal axis (labeled Extra Cost) means that there is a very good chance that the new treatment is more expensive than usual care, represented by '0' (since ΔC is shown to be >0 in all instances). In contrast, some of the ΔE and ΔC couplets are on the left side of the vertical dashed line (16.7% to be exact) and some are on the right side (83.3%). This means it is not 100% clear that the new treatment is more effective. The odds are good (5 to 1), but there is still a 17% chance that the healthcare payer who funds this treatment will be paying more for a less effective treatment than usual care. This is because the estimates are spread between the north-east and north-west quadrants, signifying uncertainty in effect.

### 1.2.4. Cost-Effectiveness Acceptability Curve (CEAC)

Importantly, the likelihood that the new drug is cost-effective is likely less than 83.3%. This is because the probability a new treatment is cost-effective is related to the value of the extra gain in patient outcome in relation to the extra cost. A new treatment could show evidence of being more effective but be priced in such a way that the extra cost makes the investment an unattractive one. For example, one of the 1000 ΔE and ΔC couplets is plotted at (0.0128, 5516.93), suggesting that the extra effect is positive (0.0128 > 0) for a modest extra cost around USD 5517; however, the ICER is not very economically attractive at over USD 430,000 (i.e., 5517/0.0128). A decision maker willing to pay USD 400,000 for an additional year of life would value the 0.0128 gain in life years (i.e., USD 5120) to be

less than the extra cost to obtain it (i.e., USD 5517). This means that even if most/all of the uncertainty suggests that the new treatment is more effective and more costly, there is not a 100% probability the new treatment is cost-effective. One must compare the $\Delta E$ and $\Delta C$ couplets in Figure 1C to the decision maker's unknown willingness to pay (WTP) for an additional unit of outcome.

The cost-effectiveness acceptability curve (CEAC) accomplishes this task by varying the unknown WTP along the horizontal axis from USD 0 to something large (e.g., USD 1,000,000). Figure 2 shows a CEAC that appears to plateau around 80%. This is because 17% of the scatterplot provides evidence of a less effective new treatment with additional cost. When decision makers are not willing to pay anything for additional effectiveness (i.e., they only care about cost savings and WTP = 0), the probability that the new treatment is cost-effective is 0%. This is because 0% of the dots in Figure 1C are below the horizontal axis (indicating cost saving).

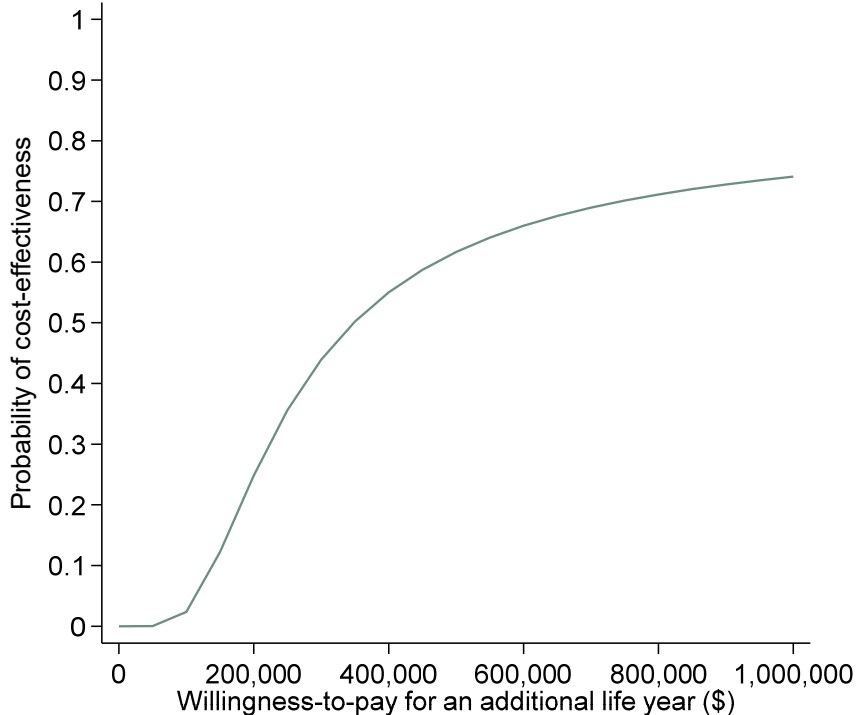

**Figure 2.** The cost-effectiveness acceptability curve (CEAC).

### 1.2.5. Incremental Net Benefit by Willingness-to-Pay plot

While the CEAC is intuitive, it does suffer from a variety of limitations [10]. For example, the CEAC does not quantify how cost effective a new treatment is, it shows only the probability that a new treatment is cost-effective. In contrast, the INB by WTP plot shows both the magnitude and uncertainty of cost effectiveness. The INB statistic and its 95% CI depend on the unknown WTP value, so plotting them against the unknown WTP value is a worthwhile activity. Figure 3 shows the results of this in an INB by WTP plot.

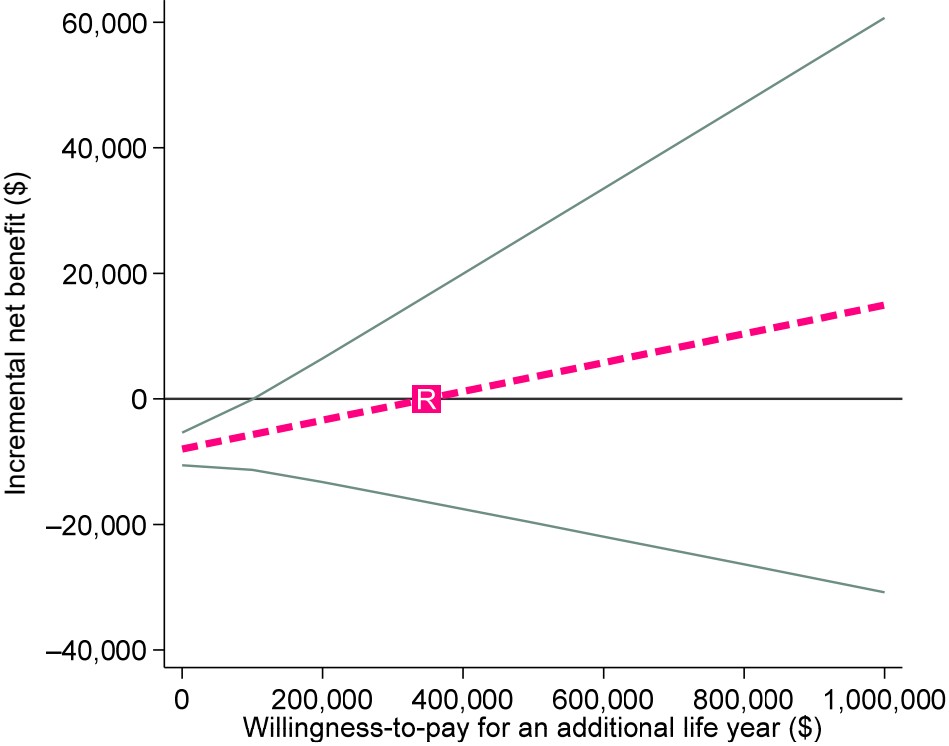

**Figure 3.** Incremental net benefit by willingness to pay.

In Figure 3, the reader can discern the ICER and INB estimates as well as the 95% CIs for both estimates [11]. The dashed line is the INB estimate, and it crosses the horizontal axis when the WTP equals the ICER estimate (marked with an R in Figure 3). The solid lines show the 95% CI for the INB, and where those lines intersect the horizontal axis reflects the ICER 95% CIs. It is clear that only one solid line intersects the horizontal axis, and this is why the 95% CI for the ICER reported in Table 1 appears problematic.

In summary, this section reviewed five common ways to characterize statistical uncertainty in a cost-effectiveness analysis: (1) the ICER 95% CI; (2) the INB 95% CI; (3) a scatterplot on the cost-effectiveness plane; (4) the CEAC; and (5) the INB by WTP plot.

## 2. Methods

### 2.1. Data

The data for our analysis of how authors communicate statistical uncertainty in real-world cost-effectiveness analysis come from a recent rapid review [4]. The included studies were identified through a primary review using the Preferred Reporting Items for Systematic Reviews and Meta-Analyses (PRISMA) [12] in July of 2022 using PubMed. Selection criteria and methodology are listed in our previous work [4]. The focus of the review was on the utilization of real-world data for cost-effectiveness analysis (CEA) of cancer treatments in Canada, with "real-world" defined as utilizing data from a cohort observational study or using individual-level person data from administrative or clinical databases. The utilization of real-world data facilitates further understanding and informing of future evidence-based initiatives in Canadian oncology. Studies containing information about extra cost and extra effect were included in the review, even if the authors did not identify the study as an economic evaluation.

### 2.2. Variable Creation

Our team extracted how authors of the 22 included studies communicated uncertainty about estimates of ΔC, ΔE, and the incremental cost-effectiveness ratio (ICER). We counted the frequency of use of the five common ways to characterize statistical uncertainty in a

cost-effectiveness analysis: (1) the ICER 95% confidence interval (CI); (2) the incremental net benefit (INB) 95% CI; (3) a scatterplot on the cost-effectiveness plane; (4) the cost-effectiveness acceptability curve (CEAC); and (5) the incremental net benefit (INB) by willingness-to-pay (WTP) plot. For scatterplots, we identified the quadrants involved (e.g., North-East (NE), North-West (NW), South-West (SW), South East (SE)).

### 3. Results

Of the 22 articles in our study, 77.3% shared uncertainty results characterized by at least one of the ways described in our Background section.

Table 2 illustrates the 22 studies selected for our analysis. The table also shows each study's cancer interventions evaluated, study design, uncertainty characterization, if any, and describes the quadrants populated in the scatterplots. In total, a small majority (59%) of studies utilized administrative data or retrospective datasets to inform their calculations and studies. The remainder (41%) used administrative and retrospective data to inform models for cancer interventions.

**Table 2.** Summary of selected article interventions and uncertainty.

| Author(s) | Uncertainty (Characterized by CI for ICER, CI for INB, INB by WTP, CEAC, Scatterplot) | Possible Location of Estimates |
|---|---|---|
| **Model** | | |
| Johnston et al. 2010. *Parmacoeconomics Outcomes Res.* [13] | Scatterplot | NE, SE, SW, NW NE, SE, NW |
| Hannouf et al. 2012. *BMC Cancer* [14] | CEAC, Scatterplot | NE, SEM SW, NW NE, NW |
| Hedden et al. 2012a. *Eur. J. Cancer Oxf.* [15] | CEAC, Scatterplot | NE, SE |
| Hedden et al. 2012b. *The Oncologist* [16] | CEAC, Scatterplot | NE, NW |
| Cressman et al. 2017. *J. Thorac. Oncol. Off. Publ. Int. Assoc. Study Lung Cancer* [17] | CEAC | - |
| Nazha et al. 2018a. *Drug Investig.* [18] | Scatterplot | NE |
| Parackal et al. 2020. *Can Urol. Assoc. J. J. Assoc. Urol. Can.* [19] | CEAC | - |
| Raymakers et al. 2020. *BMC Cancer.* [20] | CEAC, Scatterplot | NE, SE, NW |
| Cressman et al. 2021. *CMAJ Open.* [21] | CEAC, Scatterplot | NE, SE, SW, NW |
| **Dataset** | | |
| Cromwell et al. 2011. *J. Thorac. Oncol. Ogg. Publ. Int. Assoc. Study Lung Cancer.* [22] | CEAC, Scatterplot | NE, SE, SW, NW |
| Cromwell et al. 2012. *Lung Cancer Amst. Neth.* [23] | Scatterplot | NE, SE |
| Khor et al. 2014. *BMC Cancer* [24] | CI for ICER, CEAC, Scatterplot | NE |
| Thein et al. 2017. *Cancer Med.* [25] | CI for ICER, CI for INB, CEAC, INB by WTP | - |
| Mittmann et al. 2018. *J. Clin. Oncol Off. J. Am. Soc. Clin. Oncol.* [26] | None Stated | - |
| Nazha et al. 2018b. *Curr Oncol Tor Ont.* [27] | None Stated | - |
| Imran et al. 2019. *Eur. Thyroid. J.* [28] | None Stated | - |
| Gilbert et al. 2020. *J. Comp. Eff. Res.* [29] | None Stated | - |
| Pataky et al. 2021. *MDM Policy Pract.* [30] | CI for ICER, CI for INB, CEAC | - |
| Weymann et al. 2021. *J Community Genet.* [31] | CI for INB | - |
| Arciero et al. 2022. *JNCI Cancer Spectr.* [32] | CI for INB, CEAC, INB by WTP, Scatterplot | NW, SW NW, SW |
| Dai et al. 2022. *JAMA Oncol.* [33] | CI for ICER, CI for INB, CEAC, INB by WTP, Scatterplot | NE |
| Tesch et al. 2022. *Cancer.* [34] | None Stated | - |

Note: CI for ICER, confidence interval for the incremental cost effectiveness ratio. CI for INB, confidence interval for the incremental net benefit. INB by WTP, incremental net benefit by willingness to pay. CEAC, cost effectiveness acceptability curve. N (North), S (South), E (East), W (West), NW (North West), NE (North East), SW (South West), SE (South East).

Table 3 reports descriptive statistics about the techniques used to characterize uncertainty in the research from our review. These totals and percentages are for both modeling papers and papers that utilized datasets for their analyses. Among all papers, scatterplots and CEACs were used the most (34.3% and 31.4%, respectively). All modeling papers used a scatterplot and/or a CEAC, whereas papers analyzing datasets used a wider variety of techniques when expressing uncertainty: scatterplots (21.7%), CEACs (26.2%), and INB 95% CIs (21.7%) were commonly utilized in dataset-based studies.

**Table 3.** Uncertainty technique by study design for 22 studies.

| Technique | Model *n* (%) | Data *n* (%) | Total *n* (%) |
|---|---|---|---|
| ICER 95% CI | 0 (0) | 4 (17.4) | 4 (11.4) |
| INB 95% CI | 0 (0) | 5 (21.7) | 5 (14.3) |
| INB by WTP | 0 (0) | 3 (13.0) | 3 (8.6) |
| Scatterplot | 7 (50) | 5 (21.7) | 12 (34.3) |
| CEAC | 7 (50) | 6 (26.2) | 11 (31.4) |

Note: *n* = occurrences, where authors may have used more than one technique in their papers (explaining why the column sum of *n* can be larger than 22. CI for ICER, confidence interval for the incremental cost effectiveness ratio. CI for INB, confidence interval for the incremental net benefit. INB by WTP, incremental net benefit by willingness to pay. CEAC, cost effectiveness acceptability curve.

Figure 4 highlights the variability in the ways authors express statistical uncertainty in their papers. All model-based analyses (bottom row) presented uncertainty using either one or two ways (44% and 56%, respectively), with all model analyses providing readers with uncertainty results. In contrast, the majority of analyses of datasets (38%) did not show statistical uncertainty for cost-effectiveness statistics. However, when these types of papers did show uncertainty results, nearly 75% used more than one method (i.e., conditional probability calculated as $(8 + 15 + 15 + 8)/(15 + 8 + 15 + 15 + 8)$).

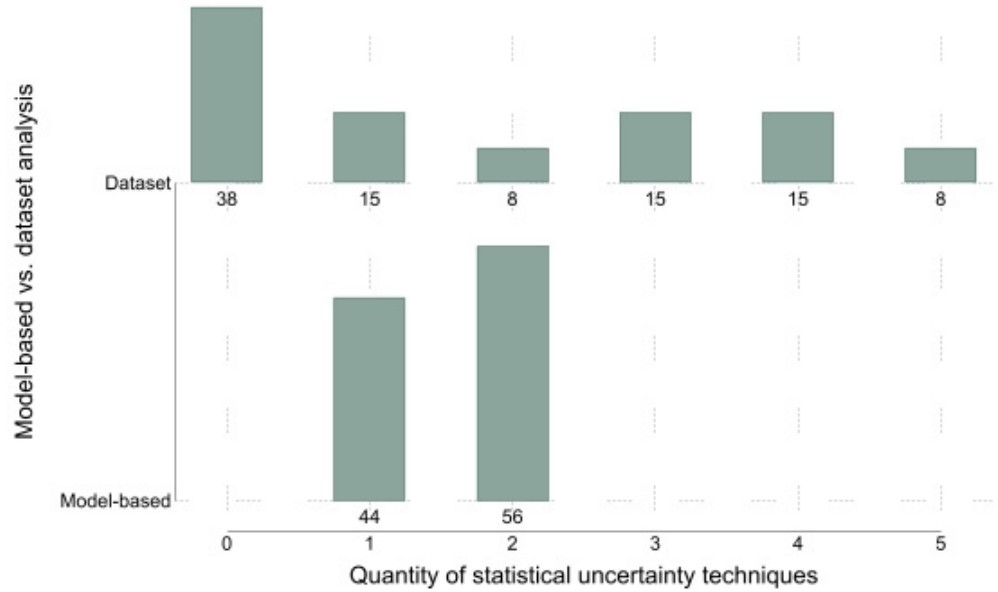

**Figure 4.** Quality of statistical uncertainty techniques used by analysis type.

The radar graph in Figure 5 is ordered by date, with the oldest article positioned at 12 o'clock and papers trending toward the present in a clockwise order. The innermost point of the plot (at the center of the circle) indicates using none of the five methods of communicating uncertainty discussed in the Background section. Moving outwards away from the center of the circle shows how many methods were used. In 2014, Khor et al. was the first author to use more than one method to show uncertainty, with three methods counted [24] We also see that only three papers [25,32,33] used four or more

methods (i.e., papers by Thein et al., Arciero et al., and also Dai et al.). Observing the pattern in the radar plot suggests there is no trend over time in the incorporation of uncertainty. However, when a member of the Pharmacoeconomics Research Unit established at Cancer Care Ontario (subsequently Ontario Health in 2019) appears as an author, the paper was observed to have larger numbers of uncertainty graphs.

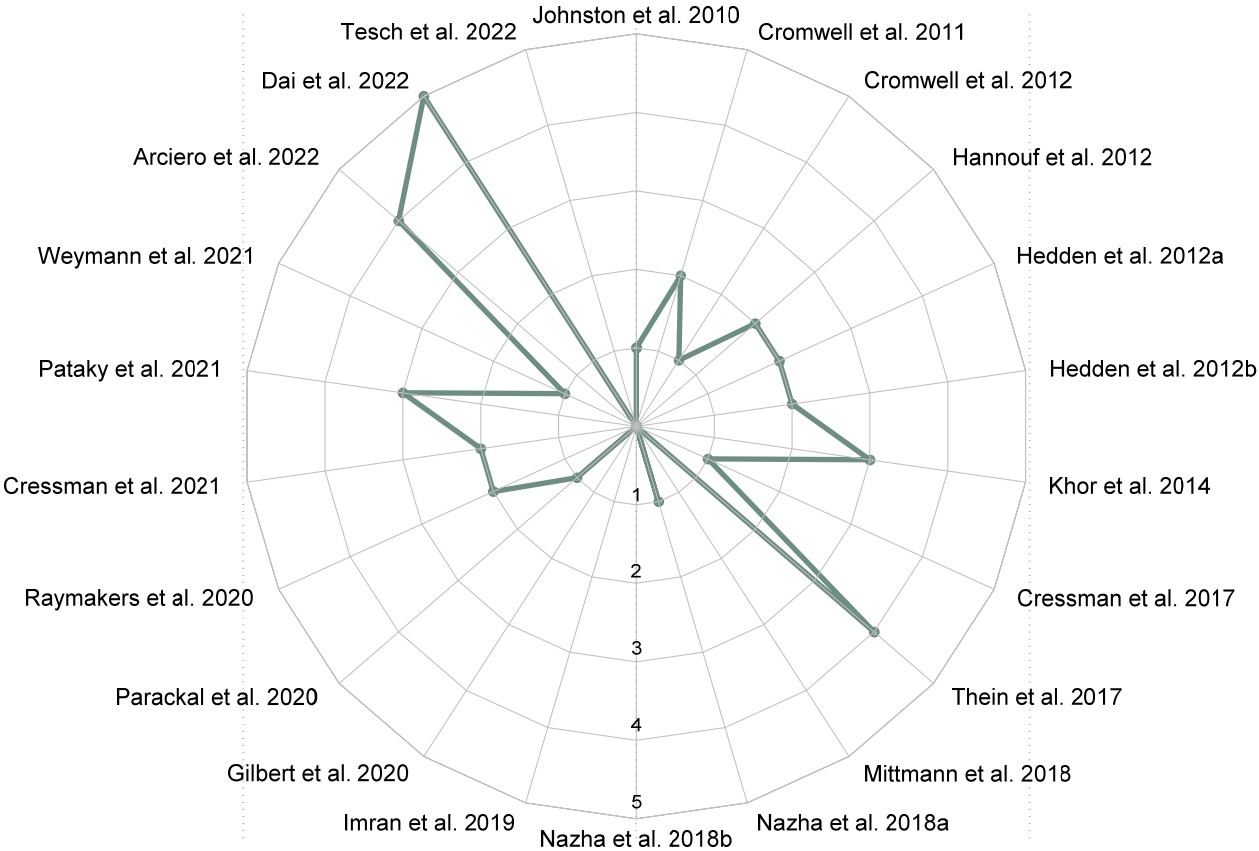

**Figure 5.** Radar graph illustrating the quantity of uncertainty methods by paper [13–34].

Through time, there is not a clear trend in how authors characterize statistical uncertainty with regards to cost effectiveness. Figure 6 is organized by year on the vertical axis and quantity of uncertainty techniques on the horizontal axis to illustrate the lack of trends in characterization over time. Additionally, studies are separated by whether they used a dataset or were model-based. Circles indicate individual studies' use of uncertainty methodologies, and triangles indicate where the average laid for the year's publications. There is much less spread in the papers that utilized models (i.e., the right panel in Figure 6). While all models used at least one form of expressing their uncertainty results, when analyses of cost-effectiveness datasets did report them, we see that they often used more than one method. This was particularly evident in the most recent papers, published in 2022, where the average number of methods was three.

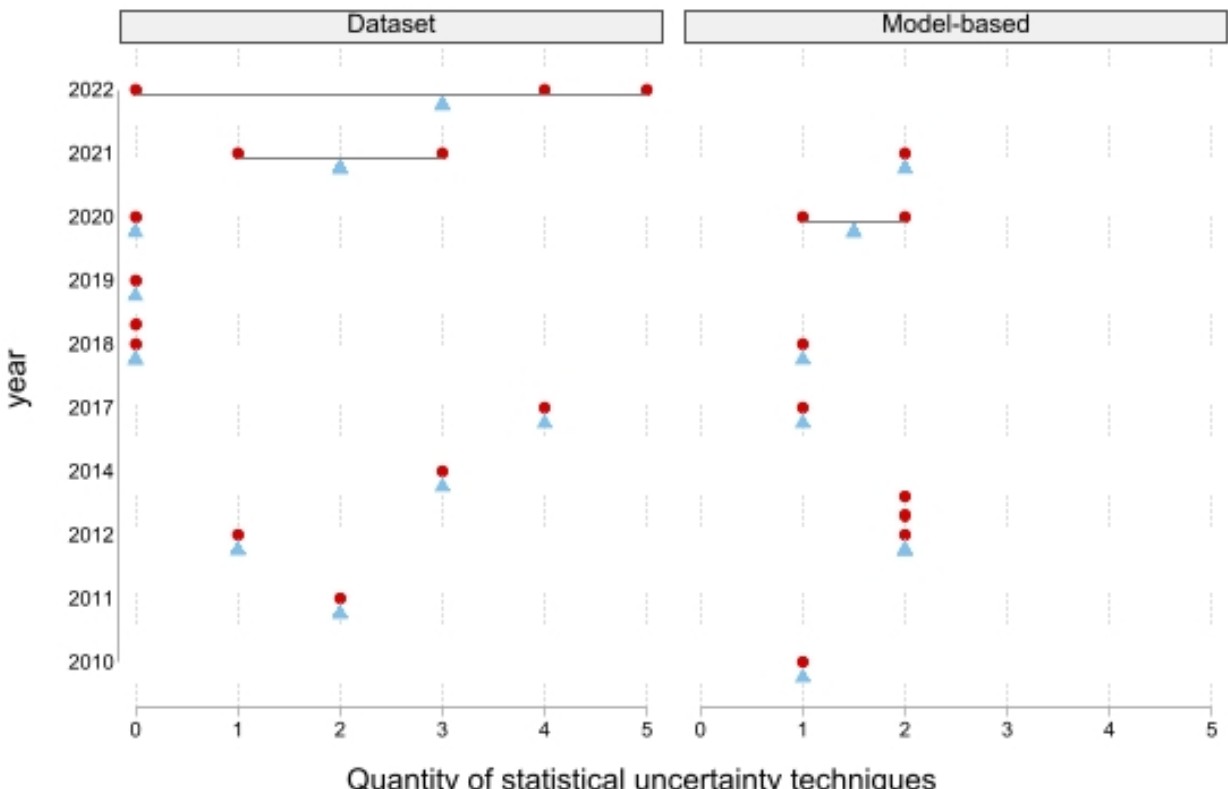

**Figure 6.** Quantity of statistical uncertainty techniques by year.

*Location of Uncertainty Findings*

Figure 7 summarizes the location and meaning of the uncertainty reported in the papers we reviewed that presented their results with a scatterplot (specifically with extra cost on the vertical axis and extra effect on the horizontal axis) [13–16,18,20–24,32,33]. Of the 15 total scatterplots [18], 9 (60%) were based on models and 6 (40%) on datasets. Only 11% of the models showed uncertainty confined to one quadrant [18], indicating high certainty in the results, while considerably more (33%) of the non-modeling papers have uncertainty confined to one quadrant [24,33]. When uncertainty was confined to one quadrant [18,24,33], it was only in the North-East quadrant, strongly suggestive of a new treatment that is both more expensive and more effective.

One in five models [14,16] showed uncertainty in two quadrants (North-East and North-West), indicating higher cost but uncertain health gain. The remainder of the studies, models, and datasets alike [13–15,20–23,32] showed uncertainty about cost savings. Those with uncertainty spread across two quadrants in the North/South direction indicate uncertainty in the extra cost of the intervention. Some were split between North-West and South-West [32], indicating less effect than usual care with uncertain cost. Others saw uncertainty across North-East and South-East [15,23], indicating more effect with uncertain cost. Results spread across three or four quadrants demonstrated high levels of uncertainty about both cost and effect [13,14,20–22].

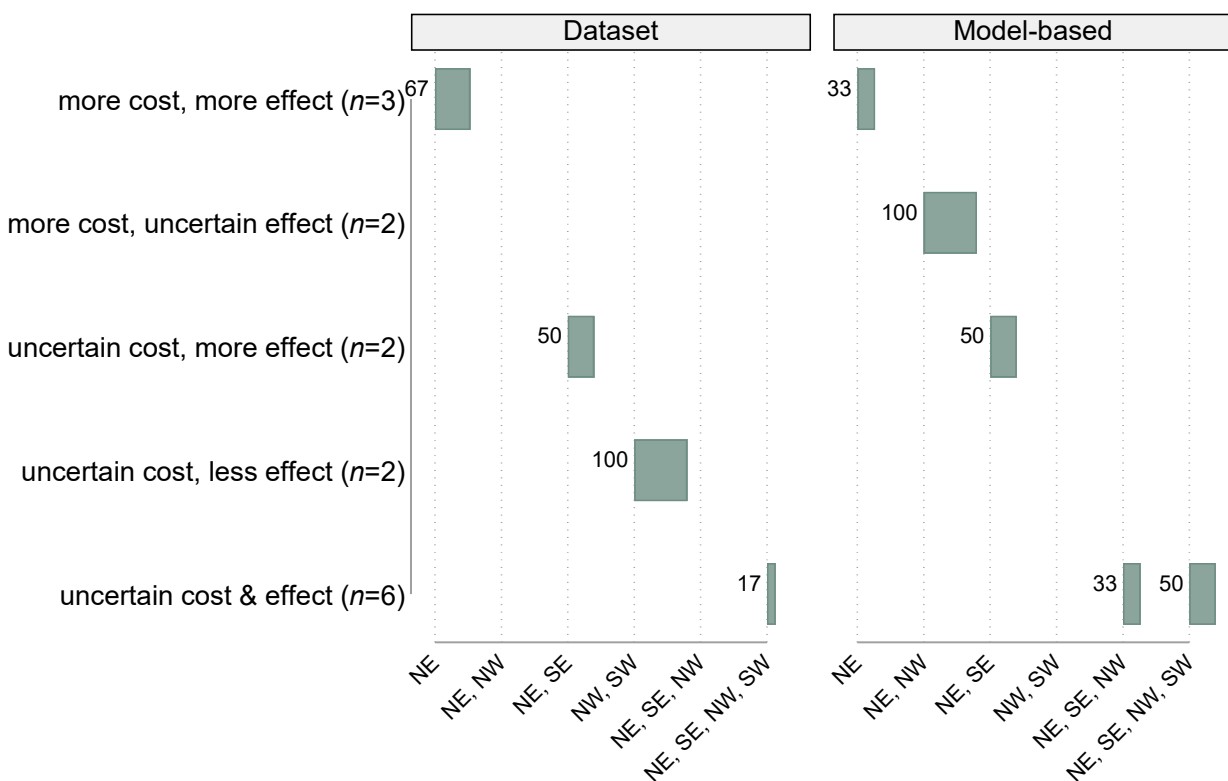

**Figure 7.** Location of uncertainty findings in scatterplots. Note: Location and meaning of uncertainty findings in scatterplots. The horizontal axis indicates location of the uncertainty in a cost-effectiveness plane and the vertical axis indicates the interpretation of the uncertainty. The numbers to the left of the "bars" are the row percentages across all (both dataset- and model-based analyses) papers. The quadrant interpretations are as follows: North-East (NE): higher cost, higher effect; NE, North-West (NW): higher cost, uncertain effect; NE, South-East (SE): higher effect, uncertain cost; NW, South-West (SW): lower effect, uncertain cost; NE, SE, NW: uncertain cost, uncertain effect; NE, SE, NW, SW: uncertain cost, uncertain effect.

## 4. Discussion

Real-world cost-effectiveness studies can provide useful information. While cost-effectiveness analysis (CEA) estimates are important, it is also important to look at the ways in which such studies account for uncertainty because this helps put their estimates in context. Whether characterizing uncertainty through 95% confidence intervals (CIs) for the incremental cost effectiveness ratio (ICER) or incremental net benefit (INB), a scatterplot on a cost-effectiveness plane, a cost-effectiveness acceptability curve (CEAC), or an INB by willingness-to-pay (WTP) plot, it is important to recognize the different and complimentary information provided by each. For example, CEACs, while informative regarding the probability of cost effectiveness, lack the nuances of uncertainty conveyed in a scatterplot. There is great benefit in reviewing both the CEAC and scatterplots to assess what is driving the cost-effectiveness claim—high cost versus high effect versus a combination of the two. In addition, scatterplots can be used to check CEAC accuracy. By taking the time to look at multiple forms of statistical uncertainty, consumers are able to critically analyze results of cost-effectiveness studies and assess the strength of evidence that new treatments are cost-effective. In addition, this observation of multiple uncertainty sources allows the reader to assess if studies have accounted for uncertainty in a meaningful way.

Bowrin et al.'s overview of the literature of cost-effectiveness analyses using real-world data noted that cost-effectiveness analysis is, "likely to be particularly valuable... to formulate appropriate treatment pathways, encourage the optimal allocation of scarce

resources, and improve aggregate patient outcomes" [35]. However, the results must also communicate information about the statistical uncertainty inherent in the analysis. This requirement is no more stringent and no less important than the need to characterize uncertainty (e.g., with 95% confidence intervals) in the seminal trials that pave the way for drugs' introductions to the marketplace. Of the papers presented in our review, just shy of one-quarter did not share uncertainty estimates, all of which used datasets to perform their analyses. It should be noted that, the papers may not have been envisioned as economic evaluations; however, reports including estimates of extra cost and extra effect, will be relevant for an audience interested in value in healthcare. These results may also indicatethat more training on interpreting uncertainty methods may be needed for those relying on results from trial-based analysis. The ICER and the INB are commonly utilized methods of estimating the efficiency of spending and value for money. These measures provide useful estimates of cost effectiveness given the data. But, these estimates are incomplete without also characterizing uncertainty. A point estimate is the best "guess" based on the evidence. However, this may not be the "truth"; the 95% CI contains the true parameter value 95% of the time. Uncertainty methods help the reader appreciate what other possibilities are congruent with the evidence. Additional estimates of the statistical uncertainty is warranted to understand the variability in costs and effectiveness and how much uncertainty may be related to heterogeneity generated by different patient populations or treatment contexts.

It is important to be able to digest information beyond an estimate, particularly when the results from economic evaluations are influencing policy decisions and have clinical implications. Considering that 40% of the scatterplots presented in this analysis have uncertainty in three or more cost-effectiveness quadrants, this represents a high level of uncertainty about exactly what the real-world value of a new treatment or intervention may be. Two-thirds of these papers conclude that the intervention under investigation was cost-effective (appropriate based on the point estimate alone), but when considered in the context of their level of uncertainty, these opinions may not enjoy strong support from the evidence. This has some interesting real-world implications—what are actual patient populations experiencing? What are the actual costs for healthcare payers? Moreover, there is a chance that the true value of a new treatment or intervention could be different from initial analyses, leading to a conclusion different from current practice or policy.

*Strengths and Limitations*

Our findings are subject to some limitations. The studies included in our original review [4] did not necessarily self-identify as cost-effectiveness studies; however, this did not preclude their inclusion in our rapid review because the estimates of extra cost and extra effect reported by the studies could be used to inform policy or practice decisions. Our selection criteria were based on whether results could be used to create estimates of extra cost and extra effect; thus, we may have missed some applicable literature. The proclivity to use this information for economic evaluation must include the same passion for clear communication of uncertainty. The lack of including the uncertainty of estimates is not necessarily a reflection of the quality of the estimate. Rather, it is a limitation of the estimates' usefulness. If we excluded the studies that treated costs and outcomes separately instead of as constituent parts of a cost-effectiveness statistic (e.g., ICER or the INB), it would have decreased our quantity of studies using no uncertainty techniques in their economic evaluation. Nevertheless, it does not diminish our main point that cost-effectiveness analysis should report estimates and characterize statistical uncertainty.

## 5. Conclusions

Real-world cost-effectiveness analysis is an important type of research to inform patients, providers, and policy makers about the actual cost and effectiveness of new

cancer treatments and interventions. However, if one only views the estimate of cost effectiveness without the context of uncertainty, decisions may not be well informed. Cost-effectiveness research may present an appealing estimate of costs and impacts of treatments, but an informed decision comes from understanding and interpreting the strength and certainty of the information presented. Further, those characterizing uncertainty must ensure proper interpretation based on such results. It is important to both characterize uncertainty in the results and communicate that uncertainty in a digestible and applicable way. Presenting and understanding uncertainty in these ways facilitates the use of real-world data.

**Author Contributions:** Conceptualization, all authors (H.K.B., A.M.G., J.S.H. and C.S.D.); Methodology, all authors (H.K.B., A.M.G., J.S.H. and C.S.D.); Data Curation, H.K.B., A.M.G. and J.S.H.; Writing—Original Draft Preparation, all authors (A.M.G., H.K.B., J.S.H. and C.S.D.); Writing—Review and Editing, all authors (A.M.G., H.K.B., J.S.H. and C.S.D.). All authors have read and agreed to the published version of the manuscript.

**Funding:** This research received no external funding.

**Institutional Review Board Statement:** Not applicable.

**Informed Consent Statement:** Not applicable.

**Data Availability Statement:** The data presented in this study are available on request from the corresponding author.

**Acknowledgments:** The authors wish to thank the Division of Health Policy and Management for donations in kind.

**Conflicts of Interest:** The authors declare no conflict of interest.

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
