# Peer review of "Real-World Cost-Effectiveness Analysis: How Much Uncertainty Is in the Results?"

_curroncol, doi:10.3390/curroncol30040310_

Round 1

Reviewer 1 Report

1. Real-world cost-effectiveness analyses of new cancer treatments and interventions provide valuable evidence that can inform and potentially recalibrate healthcare decisions (.....). 1-2 bibliographic sources are required

2. Initial funding decisions for expensive cancer drugs can be based on trial data, or, more commonly, decision models based on data and assumptions informed by previous research (.....). 1-2 bibliographic sources are required

3. Whether these estimates came from administrative datasets or were from decision models, statistical estimates are still estimates and should be accompanied by methods for expressing statistical uncertainty (.....). 1-2 bibliographic sources are required

4. Our findings are subject to some limitations. Please elaborate. What limitations would we be talking about?

5. The studies included in our review are not all studies that self-identified as cost-effectiveness studies. Are you referring to the studies carried out by the authors of the manuscript? Or is it about other authors? In this case, bibliographic references are needed to validate the statements. Is this study a review or an article with original research?

6. Bowrin et al.’s overview of the literature of cost-effectiveness analyses using real-world data noted that cost-effectiveness analysis is, "likely to be particularly valuable... to formulate appropriate treatment pathways, encourage the optimal allocation of scarce resources, and improve aggregate patient outcomes.”. In the conclusions chapter, I believe it would not be helpful to insert the results of other studies. For this reason, I recommend moving to the discussion section.

7. I recommend reforming the conclusions chapter to highlight the studies’ results.

Author Response

  1. Real-world cost-effectiveness analyses of new cancer treatments and interventions provide valuable evidence that can inform and potentially recalibrate healthcare decisions(.....). 1-2 bibliographic sources are required

Thank you for your feedback. We have added citations as suggested by numbers 1-3.

  1. Initial funding decisions for expensive cancer drugs can be based on trial data, or, more commonly, decision models based on data and assumptions informed by previous research(.....). 1-2 bibliographic sources are required

Thank you for your feedback. We have added citations as suggested by numbers 1-3.

  1. Whether these estimates came from administrative datasets or were from decision models, statistical estimates are still estimates and should be accompanied by methods for expressing statistical uncertainty(.....). 1-2 bibliographic sources are required

Thank you for your feedback. We have added citations as suggested by numbers 1-3.

  1. Our findings are subject to some limitations. Please elaborate. What limitations would we be talking about?

The main limitation of concern would be inclusion of literature, we have added a sentence calling out that concern.

  1. The studies included in our review are not all studies that self-identified as cost-effectiveness studies. Are you referring to the studies carried out by the authors of the manuscript? Or is it about other authors? In this case, bibliographic references are needed to validate the statements. Is this study a review or an article with original research?

Thank you, this is addressed by clarifying that it was in the review we conducted previously (now cited) and we clarified that it was the authors of the other papers that did not necessarily specify a CEA

  1. Bowrin et al.’s overview of the literature of cost-effectiveness analyses using real-world data noted that cost-effectiveness analysis is, "likely to be particularly valuable... to formulate appropriate treatment pathways, encourage the optimal allocation of scarce resources, and improve aggregate patient outcomes.”. In the conclusions chapter, I believe it would not be helpful to insert the results of other studies. For this reason, I recommend moving to the discussion section.

Thank you for this suggestion! We moved this portion of the conclusion to the discussion section and added more information to the conclusion.

  1. I recommend reforming the conclusions chapter to highlight the studies’ results.

Thank you for this suggestion.  We have re-crafted the Conclusion section.

Reviewer 2 Report

There is a typo in the last paragraph of the introduction. Tables 1, 2, and 3 need improved legends explaining acronyms rather than in body of table, e.g. co-ordinates in Table 2 should be abbreviated in body of table and explained in legend. The discussion and maybe introduction needs a explanation of why precision oncology treatments in particular benefit from real world data for cost-effectiveness analysis due to the small sub-populations analysed in clinical trials. 

Author Response

There is a typo in the last paragraph of the introduction.

Thank you! Great catch. We have edited this issue.

Tables 1, 2, and 3 need improved legends explaining acronyms rather than in body of table, e.g. co-ordinates in Table 2 should be abbreviated in body of table and explained in legend.

We have expanded in all the legends and we adjusted Table 2 to include abbreviations in the table and detail in the legend.

The discussion and maybe introduction needs a explanation of why precision oncology treatments in particular benefit from real world data for cost-effectiveness analysis due to the small sub-populations analysed in clinical trials. 

Thank you. This has been added to the first paragraph of the introduction and it is important context to frame our paper.

Reviewer 3 Report

Thank you for giving me the opportunity to read an interesting study.

The scatterplot approach to showing ICER uncertainty on a cost-effectiveness plane is easy to understand, and the sentence below is persuasive. There is still a 17% chance that the healthcare payer who funds this treatment will be paying more for a less effective treatment than usual care. However, this manuscript lacks important information conclusively.

Major comments:

[1] Please use the flow chart and address the selection criteria of your review. Key selection criteria are important for most readers because the authors wrote that the studies included in our review are not all studies that self-identified as cost-effectiveness studies.

[2] Readers may want to know the priority among five ways the authors identified to characterize statistical uncertainty. Which of five ways should the researcher who analyzed ICER show?

Minor comment: 

Papers shown in the references do not meet with the citation number in the text. For example, does the citation number 31 indicate Bowrin et al.?

Bowrin et al.’s overview of the literature of cost-effective-ness analyses using real-world data noted that cost-effectiveness analysis is, "likely to be particularly valuable... to formulate appropriate treatment pathways, encourage the optimal allocation of scarce resources, and improve aggregate patient outcomes.”.31

31. Hedden, L.; O’Reilly, S.; Lohrisch, C.; Chia, S.; Speers, C.; Kovacic, L.; Taylor, S.; Peacock, S. Assessing the Real-World Cost-Effectiveness of Adjuvant Trastuzumab in HER-2/Neu Positive Breast Cancer. The Oncologist 2012, 17, 164–171, doi:10.1634/the-oncologist.2011-0379.

Author Response

Thank you for giving me the opportunity to read an interesting study.

The scatterplot approach to showing ICER uncertainty on a cost-effectiveness plane is easy to understand, and the sentence below is persuasive. There is still a 17% chance that the healthcare payer who funds this treatment will be paying more for a less effective treatment than usual care. However, this manuscript lacks important information conclusively.

We appreciate both your positive and constructive feedback. We have worked to incorporate your comments. We responded to them below in red and in comments in the paper.

Major comments:

[1] Please use the flow chart and address the selection criteria of your review. Key selection criteria are important for most readers because the authors wrote that the studies included in our review are not all studies that self-identified as cost-effectiveness studies.

So as to not re-state our previous methods section, we included an additional sentence directing the reader to our previous paper and added a sentence emphasizing the availability of those in-depth methods.

[2] Readers may want to know the priority among five ways the authors identified to characterize statistical uncertainty. Which of five ways should the researcher who analyzed ICER show?

Thank you for this meaningful comment. We feel as though this is addressed here, stressing that none are favorited but all contribute meaningful information and even better when presented in tandem with one another. We have also pasted the section below:

Discussion: “... Whether characterizing uncertainty through 95% confidence intervals (CIs) for the incremental cost effectiveness ratio (ICER) or incremental net benefit (INB), a scatterplot on a cost-effectiveness plane, a cost-effectiveness acceptability curve (CEAC), or an INB by willingness to pay (WTP) plot, it is important to recognize the different and complimentary information provided by each. For example, CEACs, while informative regarding probability of cost-effectiveness, lack the nuances of uncertainty conveyed in a scatterplot. There is great benefit in reviewing both the CEAC and scatterplots to assess what is driving the cost-effectiveness claim – high cost versus high effect versus a combination of the two. In addition, scatterplots can be used to check CEAC accuracy. By taking the time to look at multiple forms of statistical uncertainty, consumers are able to critically analyze results of cost-effectiveness studies and assess the strength of evidence that new treatments are cost-effective. In addition, this observation of multiple uncertainty sources allows the reader to assess if studies have accounted for uncertainty in a meaningful way.”

Minor comment: 

Papers shown in the references do not meet with the citation number in the text. For example, does the citation number 31 indicate Bowrin et al.?

Bowrin et al.’s overview of the literature of cost-effective-ness analyses using real-world data noted that cost-effectiveness analysis is, "likely to be particularly valuable... to formulate appropriate treatment pathways, encourage the optimal allocation of scarce resources, and improve aggregate patient outcomes.”.31

  1. Hedden, L.; O’Reilly, S.; Lohrisch, C.; Chia, S.; Speers, C.; Kovacic, L.; Taylor, S.; Peacock, S. Assessing the Real-World Cost-Effectiveness of Adjuvant Trastuzumab in HER-2/Neu Positive Breast Cancer. The Oncologist 2012, 17, 164–171, doi:10.1634/the-oncologist.2011-0379.

Thank you for pointing this out. We have updated and reformatted the references using our citation software and manually checked the update.